# Bone Mineral Reference Values for Athletes 11 to 20 Years of Age

**DOI:** 10.3390/ijerph17144930

**Published:** 2020-07-08

**Authors:** Irina Kalabiska, Annamária Zsakai, Robert M. Malina, Tamas Szabo

**Affiliations:** 1Research Center for Sport Physiology, University of Physical Education, Alkotas u. 44, 1123 Budapest, Hungary; kalabiskai@gmail.com (I.K.); szabo.tamas@tf.hu (T.S.); 2Department of Biological Anthropology, Eotvos Lorand University, Pazmany P. s. 1/c, 1117 Budapest, Hungary; 3Department of Kinesiology and Health Education, University of Texas, Austin, TX 78712, USA; rmalina@1skyconnect.net

**Keywords:** DEXA, bone mineral, bone development, youth athletes

## Abstract

Objectives. Training for sport is associated with the development of bone minerals, and the need for reference data based on athletes is often indicated. The purpose of this study was to develop a reference for bone mineral density (BMD) and content (BMC) specific for youth athletes of both sexes participating in several sports. Methods DEXA (dual energy X-ray absorptiometry) was used for total body measurements of bone minerals in 1385 athletes 11 to 20 years, 1019 males and 366 females. The athletes were training in several sports at Hungarian academies. Reference values for total bone mineral density and bone mineral content, and also BMD excluding the head (total body less head, TBLH) were developed using the LMS chartmaker pro version 2.3. Results. The centile distributions for BMD and BMC of the athletes differed significantly from those of the age- and sex-specific references for the general population. The youth athletes had higher BMD and BMC than those of the reference for the general population. Conclusion. The potential utility of the DEXA reference for male and female youth athletes may assist in monitoring changes in the BMC and BMD associated with normal growth and maturation, and perhaps more importantly, may be useful in monitoring changes specific to different phases of sport-specific training protocols.

## 1. Introduction

Concern for bone health, specifically bone mineral density (BMD) and bone mineral content (BMC), is a major concern given age and gender variation [1,2]. In this context, reference data for a population are routinely used for screening purposes; as such, they are a reference for comparison [1,3,4]. Given changes in BMD and BMC during the course of growth and maturation, there is increased interest in the bone health of children and youth, especially in the context of beneficial effects of regular physical activity on BMC [5,6,7]. Evidence is also consistent in showing beneficial effects of systematic training for sport on BMC [8,9]. Mechanical loading associated with regular training has a significant influence on skeletal development and bone maintenance. Regular intensive exercise with considerable mechanical loading in athletes is associated with an increase in absolute and relative bone dimensions and structural parameters. In contrast, inadequate bone development per se and some practices associated with specific sports, e.g., extreme weight control measures and disordered eating, can negatively influence performance and can increase the risk for bone stress injuries. Understanding bone development in young athletes may inform training practices, leading to success in sport, facilitate the diagnosis of structural abnormalities in bone, and contribute to the prevention of skeletal injuries [10,11,12].

The bone health of athletes, specifically BMD and BMC status, is generally evaluated relative to reference data for the general population [13,14,15]. Such a reference may have limitations with athletes given the selectivity of sport in general, sport-specific training demands, and dietary pressures associated with specific sports [16,17]. Moreover, measures of BMD and BMC among athletes often fall outside normal ranges for the general population [18,19,20].

DEXA (dual energy X-ray absorptiometry) technology provides measures of total body bone area (cm^2^), BMC (g), and BMD (g/cm^2^) [21]. However, evaluation of BMC and BMD requires appropriate reference data that allow for chronological age, body size, pubertal status, ethnicity, and sex, and perhaps demands of specific sports [22,23,24]. The preferred indicator of BMD is the total body, excluding the head [25]. Evidence suggests that age, height, and weight were better predictors of total body BMD excluding the head (total body less head (TBLH) BMD) in contrast to total body BMD [26]. Use of the subtotal BMD result, excluding the head region, is preferred as the skull does not develop in a proportionally similar manner to body mass and to the weight of other organs in children and adolescents. In addition, the head constitutes a large portion of the total body bone mass but changes little with growth, activity, or disease. Thus, including the skull may mask gains or losses at other skeletal sites [21].

The purpose of this study was to develop reference values for BMC, BMD, and TBLH BMD based on DEXA in a sample of Hungarian youth athletes of both sexes.

## 2. Material and Methods

The project was approved by the Ethics Committee of the University of Physical Education in Budapest, Hungary. Parents of athletes <18 years and also the athletes were informed of the details of the project; both provided written informed consent. Details of the project were also given to older athletes who also provided informed written consent.

Participants were 1385 athletes, 1019 males and 366 females, 11–20 years of age, who volunteered to participate in this cross-sectional study. The athletes represented several Hungarian sport academies, primarily basketball, football, and handball with smaller numbers for ice hockey and several individual sports including pentathlon, rhythmic gymnastics, swimming, athletics, fencing, kayak, canoe, rowing, wrestling, karate, and weight-lifting. All of the participants in the study were considered elite and most began training at 6–7 years of age. The athletes trained daily for approximately two hours per day through most of the year and had at least one competition per week. The respective academies delegated the athletes for the body structural and DEXA examinations. Exclusion criteria for DEXA scans included the lack of written consent, body weight >130 kg, height >200 cm, pregnancy, non-removable objects (e.g., prostheses or implants) in the past one-half year, and inability of an athlete to attain correct position and/or to remain motionless during the scan.

The research was conducted between September 2015 and March 2019. Whole body bone mineral density (BMD), bone mineral content (BMC), and total body less head (TBLH BMD) were measured with a GE Lunar Prodigy dual-energy X-ray scanner. The scanner was located in the Research Center for Sport Physiology, University of Physical Education, Budapest, Hungary, and the first Author (IK) made the DEXA measurements. The data were processed with enCORE Version 16 software. The Lunar Prodigy reference data were based on an international sample of healthy children and adults from the general population in several regions of the world in the 1990s and early 2000s. The sample was free of people with chronic diseases affecting bone structure and development and those taking bone-altering medications. The enCORE software of the scanner permits comparison of a subject’s results to a selected reference population considering ethnicity (Black, White, Asian, or Hispanic; White was chosen in the case of the Hungarian young athletes). The athletes were grouped into single year chronological age groups with the whole year as the midpoint, i.e., 11 years = 10.50 to 11.49, 12 years = 11.50 to 12.49, etc. Sample sizes and descriptive statistics for age, height, and weight of male and female athletes are summarized by age groups in Table 1 and Table 2, respectively.

Age- and sex-specific means and standard deviations, and selected percentiles (10th, 25th, 50th, 75th, 90th) were calculated for BMD, BMC, and TBLH BMD using the LMS chart maker pro version 2.3. The bone mineral parameters of each athlete were converted to z-scores relative to age- and sex-specific reference values specified by the Lunar Prodigy type dual-energy X-ray scanner manual. The distributions of BMD and BMC parameters in the athletes were compared to standard reference centile distributions by using individual z-scores of bone mineral parameters in young athletes on the basis of the reference centile distribution (reference L, M, S parameters). Single sample t-tests were used to evaluate the distribution of z-scores in each age-group of males and females, i.e., to compare age and sex-specific z-scores for BMD and BMC of the athletes to the Lunar Prodigy reference for youth. The normality of z-scores for BMD and BMC was tested by the Kruskal–Wallis test. Hypotheses were tested at a 5% level of random error.

### Ethical Approval Information

The Research Ethics Committee of the University of Physical Education (Budapest, Hungary) approved the study (ID of approval: TE-KEB/No42/2019). The investigations were carried out following the rules of the Declaration of Helsinki of 1975 (https://www.wma.net/what-we-do/medical-ethics/declaration-of-helsinki/), revised in 2013.

## 3. Results

Sex differences in the selected bone mineral parameters are apparent at 14 years and older; total BMD, TBLH BMD, and BMC of male athletes are greater than corresponding values in their female age-peers except for total BMD (tBMD) at 13, 14, 19, and 20 years, and TBLH BMD at 20 years. At each indicated age, the BMD parameters do not differ between boys and girls (Table 3).

The total BMD of male athletes is considerably higher than that of the age-specific reference for males (Figure 1, *p* < 0.001 in each age-group). The median BMD curve exceeds the 90th percentiles of the reference. Although the data are cross-sectional, the adolescent increase in BMD occurs at a somewhat later age among the athletes.

The corresponding trend for total BMD among female athletes also indicates higher BMD (Figure 1, *p* < 0.001 in all age groups except 20 years, *p* = 0.02). Of interest, the 25th percentile for female athletes is higher than the 90th percentile of the reference. Nevertheless, the percentiles for female athletes parallel those of the reference.

Percentiles for BMD excluding the head (TBLH BMD in male athletes are higher than those of the reference beginning at 11 years (Figure 2, *p* < 0.001 in all age groups), while the 25th percentiles of the athletes approximate the 90th percentiles of the reference. In contrast to total body BMD, the adolescent spurt in TBLH BMD appears to be somewhat earlier in athletes compared to that of the reference.

The trend in TBLH BMD percentiles of female athletes is similar to that in males (Figure 2, *p* < 0.001 in all age groups except 13 years, *p* = 0.04). The cross-sectional data also suggest a more intense adolescent spurt in female athletes.

Percentiles for BMC among male athletes are also consistently higher than those of the reference (Figure 3, *p* < 0.001 in all age groups). The medians for athletes approximate the 90th percentiles of the reference across the age range except at 12 years when the 25th percentile of the athletes approximated the 90th percentile of reference.

The corresponding trends for BMC of female athletes generally parallel those of the reference percentiles from 14 years on, but are significantly higher than those of the reference (Figure 3, *p* < 0.001 in all age groups except 13 years, *p* = 0.04).

Examples of the application of the reference percentiles for athletes in the evaluation of individual BMDs are shown in Figure 4. The athletes were selected from a longitudinal study of bone structure in youth athletes. In the male athlete (Figure 4, left), the BMD of the athlete is higher than the 90th percentile of the general population at each observation, but relative to the reference for athletes, the BMD shifts from the median at observation 1 to the 75th percentile at observation 2, suggesting that BMD continued to increase between 17 and 19 years. In the example of the female athlete (Figure 4, right), BMD values are higher than the 90th percentile of the general population reference across the age interval considered, but are below the reference median for athletes and gradually decline over time to <25th percentile of the athlete reference by 20 years of age. Such a decline in BMD relative to the athlete reference suggests the need for attention from the trainers as to potential factors associated with the decline.

## 4. Overview

Comparison of the percentiles of DEXA measurements of total BMD, BMD TBLH, and BMC of athletes 11–20 years in several sports (Appendix A, Table A1, Table A2 and Table A3) with the corresponding Lunar Prodigy reference percentiles highlighted the contrast in bone development between athletes and the reference, and by inference suggested a need for reference values specific for youth athletes. References data for DEXA bone parameters for athletes are limited [27]. As such, the reference values for athletes in the present study may aid in the understanding of bone development in actively training youth athletes.

Relative to age- and sex-peers in the general population [28], BMD and BMC are better developed among the youth athletes. The results were consistent with earlier studies of youth athletes, and highlighted the importance of regular physical activity associated with specific sports on bone structure, density, and morphology [12,29,30,31]. Intensive physical activity associated with systematic training was associated with increased BMC and BMD. The data, however, were based on a combined sample of athletes in a variety of sports, and in the context of the literature, highlight the need to evaluate BMD and BMC in specific sports and also in sports characterized by variation in impact forces, e.g., high-impact (gymnastics, judo, karate, volleyball, etc.), odd-impact (soccer, basketball, step-aerobics, etc.), and non-impact or negative-impact (cycling, swimming, water polo, etc.) stresses. In addition to BMD and BMC, it is also important to consider variation in bone geometry in anatomic regions specific to variation in loading patterns by sport.

## 5. Limitations of the Study

Given the limited number of female athletes, the analysis of sex differences in BMD and BMC should be evaluated with caution. The lack of an indicator of maturity status in the younger athletes is also a limitation.

## 6. Summary

Reference values for DEXA measures of BMD, TBLH BMD, and BMC were developed for a relatively large sample of youth athletes 11–20 years of age.Compared to reference values for the general population (White ethnicity), BMD and BMC of the youth athletes were better developed.By inference, comparison of DEXA observations of athletes with reference values for the general population must be done with care to avoid potential misinterpretations.

## Figures and Tables

**Figure 1 ijerph-17-04930-f001:**
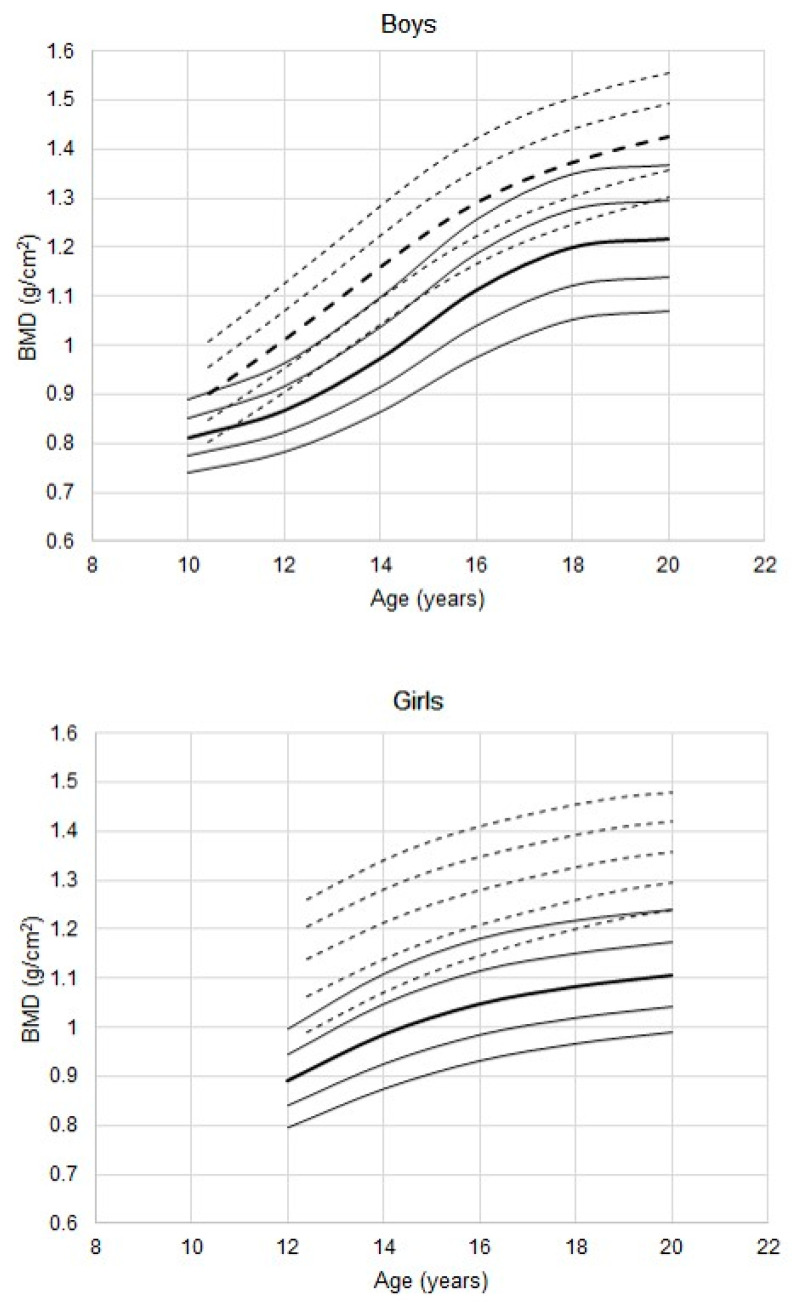
Total bone mineral density (BMD) in youth male and female athletes 11 to 20 years of age; percentiles (- - -) for athletes estimated with the LMS method, are plotted relative to the DEXA (dual energy X-ray absorptiometry) reference (—) percentiles (10th, 25th, 50th, 75th, 90th; the 50th percentiles are in bold).

**Figure 2 ijerph-17-04930-f002:**
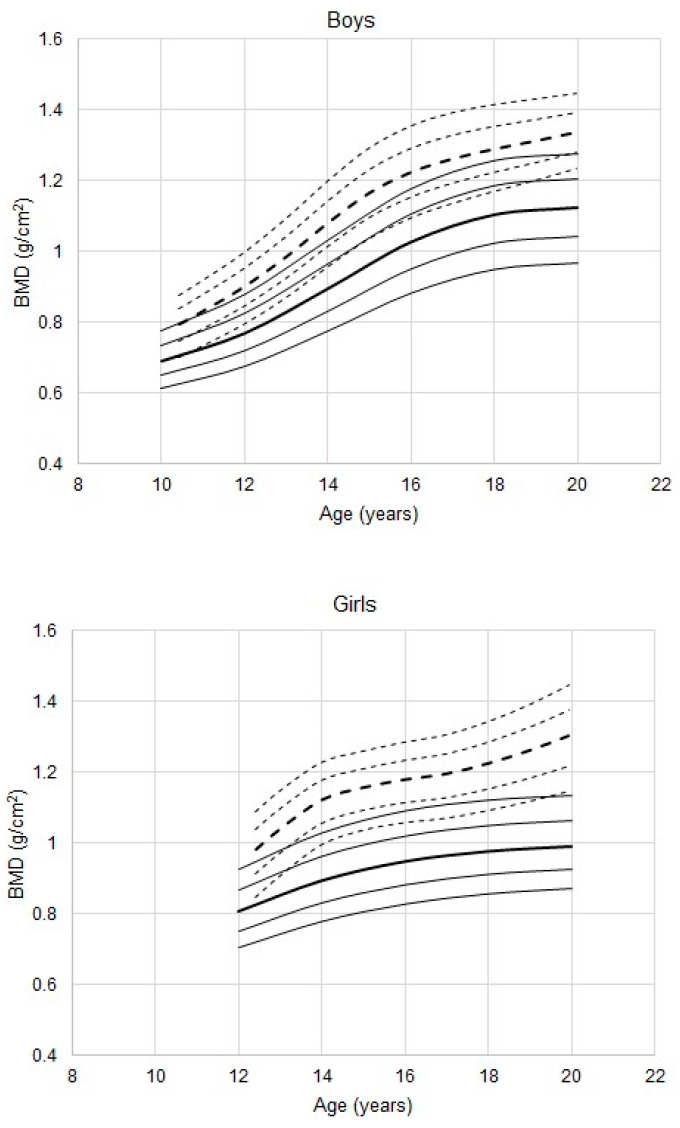
Total body less head (TBLH) BMD in youth male and female athletes 11 to 20 years of age; percentiles (- - -) for athletes, estimated with the LMS method, are plotted relative to the DEXA reference (—) percentiles (10th, 25th, 50th, 75th, 90th; the 50th percentiles are in bold).

**Figure 3 ijerph-17-04930-f003:**
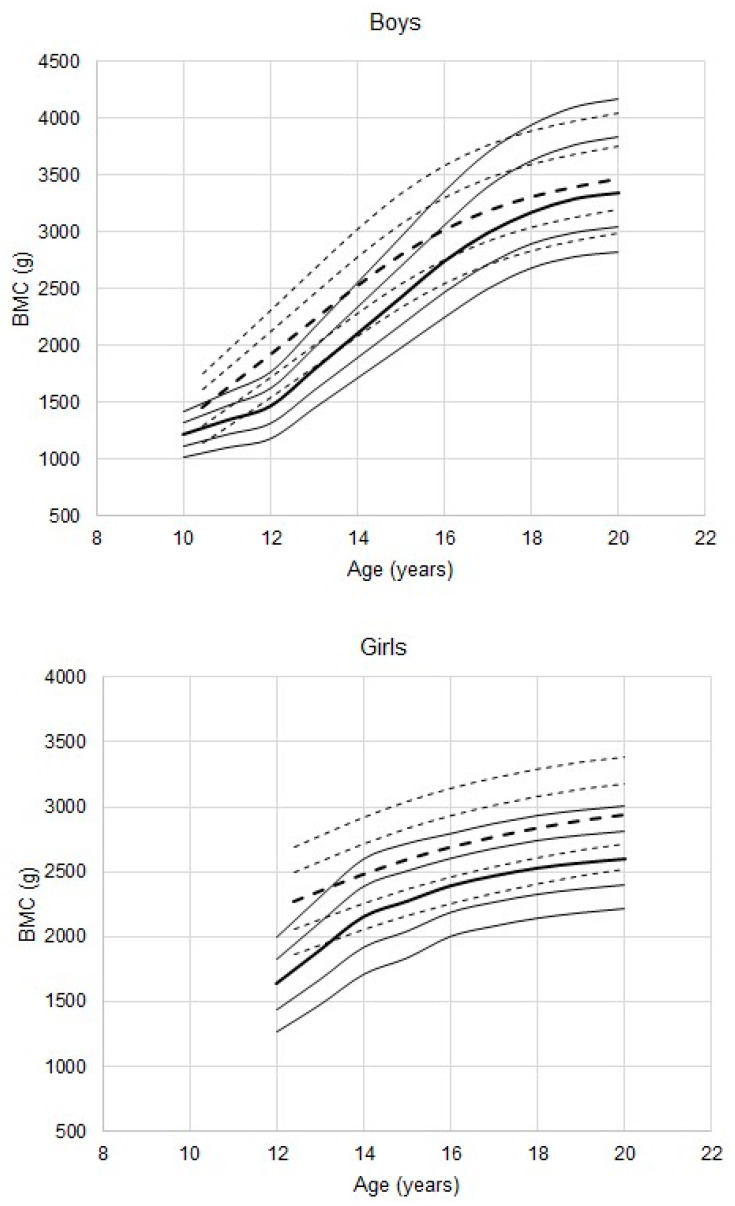
Total BMC in youth male and female athletes 11 to 20 years of age; percentiles (- - -) for athletes, estimated with the LMS method, are plotted relative to the DEXA reference (—) percentiles (10th, 25th, 50th, 75th, 90th; the 50th percentiles are in bold).

**Figure 4 ijerph-17-04930-f004:**
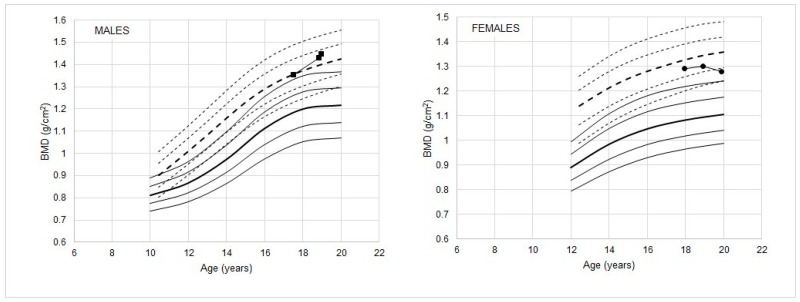
BMD of two athletes, a female handball player and a male football player, is shown relative to the percentiles (- - -) for athletes and the DEXA reference (—) percentiles (10th, 25th, 50th, 75th, 90th; the 50th percentiles are in bold).

**Table 1 ijerph-17-04930-t001:** Sample sizes and descriptive statistics (means (M) and standard deviations (SD)) for age, height, and weight of male athletes by age group.

Age Group	Sample	Age		Height		Weight	
(years)	*n*	M	SD	M	SD	M	SD
11	25	11.56	0.28	157.62	6.28	43.19	6.23
12	28	12.63	0.26	166.89	9.05	50.75	12.72
13	63	13.55	0.28	173.04	10.57	57.65	11.55
14	98	14.57	0.30	177.09	8.86	63.98	10.67
15	200	15.50	0.27	180.08	8.79	69.12	11.94
16	237	16.48	0.29	181.87	9.19	71.27	9.48
17	174	17.49	0.28	184.74	9.65	76.65	11.09
18	123	18.43	0.29	182.27	8.21	76.52	12.19
19	42	19.53	0.32	180.76	7.39	76.13	6.50
20	29	20.34	0.27	177.40	6.00	78.91	11.95

**Table 2 ijerph-17-04930-t002:** Sample sizes and descriptive statistics (means (M) and standard deviations (SD)) for age, height, and weight of female athletes by age group.

Age Group	Sample	Age		Height		Weight	
(years)	*n*	M	SD	M	SD	M	SD
13	25	13.60	0.23	168.24	8.86	57.27	10.32
14	79	14.51	0.24	171.47	7.06	63.76	9.36
15	86	15.55	0.30	173.46	7.26	67.11	11.36
16	49	16.54	0.30	172.70	7.08	65.35	8.96
17	63	17.37	0.32	173.00	7.01	66.98	12.78
18	27	18.52	0.32	174.46	8.14	66.10	7.31
19	27	19.43	0.24	174.16	6.69	70.14	9.64
20	10	20.65	0.24	167.38	8.14	63.53	7.16

**Table 3 ijerph-17-04930-t003:** Significance levels for sex differences in BMD and bone mineral content (BMC) parameters among athletes (significant values in italics).

Age (years)	Total BMD (g/cm^2^)	TBLH BMD (g/cm^2^)	BMC (g)
12	0.378	0.669	0.505
13	0.421	0.468	0.806
14	*<0.001*	*<0.001*	*0.009*
15	0.157	*0.009*	*<0.001*
16	0.127	*0.012*	*<0.001*
17	*0.012*	*<0.001*	*<0.001*
18	*0.015*	*<0.001*	*<0.001*
19	0.384	*0.024*	*0.002*
20	0.557	0.773	*0.004*

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
