# Peer review of "Bone Mineral Reference Values for Athletes 11 to 20 Years of Age"

_ijerph, 2020, doi:10.3390/ijerph17144930_

Round 1

Reviewer 1 Report

Introduction:

More information could be provided on the introduction why does this specific population need to be studied?

Line 30: Need a reference

Line 32: Compared to what?

Results: Were there differences by sex? What about differences in men and women, the sample sizes are not even, that is a huge deal in my opinion.

Author Response

The Authors would like to express sincere thanks to the Reviewer for careful reading and suggestion for improvement in the paper.

Comment 1: More information could be provided on the introduction why does this specific population need to be studied?

Response: The Authors thank this question of the Reviewers. The manuscript was completed in the Introduction section with followings:

“Mechanical loading associated with regular training has a significant influence on the skeletal development and bone maintenance. Regular intensive exercise with considerable mechanical loading in athletes is associated with an increase in absolute and relative bone dimensions and structural parameters. In contrast, inadequate bone development per se and some practices associated with specific sports, e.g., extreme weight control measures and disordered eating, can negatively influence performance and can increase the risk for bone stress injuries. Understanding bone development in young athletes may inform training practices leading to success in sport and also facilitate the diagnosis structural abnormalities in bone and contribute to the prevention of skeletal injuries (Carbuhn et al. 2010, Tenforde et al. 2017).”

The References section was completed with the following cited items:

Tenforde, A.S., Carlson, J.L., Chang, A., Sainani, K.L., Shultz, R., Kim, J.H., Cutti, P., Golden, N.H., Fredericson, M. Association of the female athlete triad risk assessment stratification to the development of bone stress injuries in collegiate athletes. Am J Sport Med 2017, 45(2), 302–310. doi: https://doi.org/10.1177%2F0363546516676262

Carbuhn, A.F., Fernandez, T.E., Bragg, A.F., Green, J.S., Crouse, S.F. Sport and training influence bone and body composition in women collegiate athletes. J Strength Condition Res 2010, 24(7), 1710–1717. doi: https://doi.org/10.1519/JSC.0b013e3181d09eb3

Comment 2: Line 30: Need a reference

Reference: The Authors thank this comment of the Reviewer. The manuscript was completed with the following reference:

Elhakeem, A., Frysz, M., Tilling, K., Tobias, J.H., Lawlor, D.A. Association between age at puberty and bone accrual from 10 to 25 years of age. JAMA Network Open 2019, 2(8), e198918-e198918

Comment 3: Line 32: Compared to what?

Response: The Authors thank this question of the Reviewer. The manuscript was corrected as follows:

“Concern for bone health, specifically bone mineral density (BMD) and bone mineral content (BMC), is a major concern given age and gender variation. In this context, reference data for a population are routinely used for screening purposes; as such, they are a reference for bone health evaluation.”

Comment 4: Results: Were there differences by sex? What about differences in men and women, the sample sizes are not even, that is a huge deal in my opinion.

Response: The Authors thank this question of the Reviewer as well. The manuscript was completed with the basic statistical comparison of male and female athletes bone structural parameters as follows:

“Sex differences in the selected bone mineral parameters are apparent at 14 years and older; total BMD, TBLH BMD and BMC of male athletes are greater than corresponding values in their female age-peers except for tBMD at 13, 14, 19 and 20 years, and TBLH BMD at 20 years. At the indicated age, the BMD parameters do not differ between boys and girls (Table 3).”

Table 3. Significance levels for sex differences in BMD and BMC parameters among athletes (significant values in italics)

Age
(years)

total BMD (g/cm2)

TBLH BMD (g/cm2)

BMC
(g)

12

0.378

0.669

0.505

13

0.421

0.468

0.806

14

<0.001

<0.001

0.009

15

0.157

0.009

<0.001

16

0.127

0.012

<0.001

17

0.012

<0.001

<0.001

18

0.015

<0.001

<0.001

19

0.384

0.024

0.002

20

0.557

0.773

0.004

The Limitation of the study section was inserted into the manuscript, and the problem of the statistical analysis due to the not even sample sizes is mentioned in this section as follows:

“Given the limited number of female athletes, the analysis of sex differences in BMD and BMC should be evaluated with caution. The lack of an indicator of maturity status in the younger athletes is also a limitation.”

Reviewer 2 Report

In the manuscript “Bone mineral reference values for athletes 11 to 20 years of age" authors the authors try to develop references for bone mineral density (BMD) and content (BMC) specific for youth athletes. The topic is interesting and also the size of the population is adequate. I find the statistics well describe and correctly used. Thank you for giving me the opportunity to revise it. I hope my criticism will help to improve the manuscript.
I would like to highlight a few points that should be changed or added.

  1. "Lunar Prodigy reference for youth" was used as reference values to which the authors refer. It should be described more precisely where these reference values are from and for which population they were established.
  2. Are there publications that estimate BMD and BMC on the European population of young non-athletes and compare these results to the Lunar Prodigy reference values?
  3. Line 158 “compared to age- and sex-peers in the general population” - citation is needed.
  4. Formulating the final conclusions, it should be clarified for which group are the references, they are or for the Caucasian race?
  5. What is the significance of how many years they train? Has this relationship been observed?
  6. The bibliography is very extensive, the formatting is correct. But the cited articles are rather old. The latest are only 4 articles from 2017, there are no newer ones. I suggest to review the latest publications and complete the bibliography
  7. English is correct. I only have some suggestions:
    • Line 35 I suggest you use “systematic sports training” instead of “systematic training for sport”
    • When using the abbreviation TBLH BMD, introduce it on line 45, and then on line 69-70 use the shortcut itself, without the full name.
    • Line 58 double “were” – correct the sentence. Try to not use the word “subjects”, participants sound better.

In general, the work is concise, it is pleasant to read.

Author Response

The Authors would like to express sincere thanks to the Reviewer for careful reading and suggestion for improvement in the paper.

General comment: In the manuscript “Bone mineral reference values for athletes 11 to 20 years of age” the authors try to develop references for bone mineral density (BMD) and content (BMC) specific for youth athletes. The topic is interesting and also the size of the population is adequate. I find the statistics well describe and correctly used. Thank you for giving me the opportunity to revise it. I hope my criticism will help to improve the manuscript.

I would like to highlight a few points that should be changed or added.

Response: We are grateful for the general comment of the Reviewer, the replies to the suggestions and comments are presented in the order of the Reviewer’s comments.

Comment 1: "Lunar Prodigy reference for youth" was used as reference values to which the authors refer. It should be described more precisely where these reference values are from and for which population they were established.

Response: The Authors thank this comment of the Reviewer. According to the User Manual (GE Healthcare: X-ray Bone Densitometer with enCORE v18 software - User Manual) of GE Lunar Prodigy dual-energy X-ray scanner the Lunar Prodigy reference data are based on healthy people (both children and adults) from general populations of the continents (in the USA, Australia, Brazil, China, Egypt, Finland, France, Germany, Indonesia, Italy, Japan, Korea, Mexico, Middle East, Philippine, Spain, Tunisia, Turkey and UK; studied in the 1990s and the beginning of the 2000s), who did not suffer from chronic diseases affecting bone structure and development, and were not taking bone-altering medications. The enCORE software of the scanner allows comparison of a participant’s results to the selected reference population and by considering the ethnic origin of the participants (Black, White, Asian or Hispanic origin can be chosen, White origin was chosen in the case of Hungarian young athletes).

The manuscript was completed with information on the reference values as follows:

“The Lunar Prodigy reference data were based on an international sample of healthy children and adults from the general population in several regions of the world in the 1990s and early 2000s. The sample was free of chronic diseases affecting bone structure and development and did not take bone-altering medications. The enCORE software of the scanner permits comparison of a subject’s results to a selected reference population considering ethnicity (Black, White, Asian, or Hispanic, White was chosen in the case of the Hungarian young athletes).”

Comment 2: Are there publications that estimate BMD and BMC on the European population of young non-athletes and compare these results to the Lunar Prodigy reference values?

Response: The Authors thank this question of the Reviewer as well. No, there has not been published this type of information from the European population until now. BMD and BMC data of children and young adults in Europe are available only for athletes or for special subgroups of children (having chronic illnesses, obesity, etc.).

Comment 3: Line 158 “compared to age- and sex-peers in the general population” - citation is needed.

Response: The references of the general population used in the analysis were published in the GE Healthcare: X-ray Bone Densitometer with encore v18 software - User Manual (LU46000EN-2EN Revision 1 (January 2019). The References section of the manuscript was completed with the item of the Manual and cited in Line 158 of the manuscript as follows:

“Relative to age- and sex-peers in the general population, BMD and BMC are better developed among the youth athletes (GE 2019).”

GE Healthcare. X-ray Bone Densitometer with encore v18 software - User Manual. LU46000EN-2EN Revision 1 (January 2019).

Comment 4: Formulating the final conclusions, it should be clarified for which group are the references, they are or for the Caucasian race?

Response: The Authors are grateful for this comment of the Reviewer, too. The references of Caucasian (White group according to the Manual) race were used in the analysis of DEXA data of Hungarian young athletes. The manuscript was completed as follows:

“Compared to reference values for the general population (White ethnicity), BMD and BMC of the youth athletes were better developed.”

Comment 5: What is the significance of how many years they train? Has this relationship been observed?

Response: The Authors thank this question of the Reviewer. All of the participants in the study were considered elite and most began training at 6-7 years of age. The actual analysis did not concern the relationship between the training history and DEXA parameters. The next step in the study will be to analyse this relationship as well as the variation in bone structural parameters by the sport type in the studied group of Hungarian young athletes.

The manuscript was completed with the information on the participants’ training history as follows:

“All of the participants in the study were considered elite and most began training at 6-7 years of age.”

Comment 6: The bibliography is very extensive, the formatting is correct. But the cited articles are rather old. The latest are only 4 articles from 2017, there are no newer ones. I suggest to review the latest publications and complete the bibliography.

Response: The Authors thank this comment of the Reviewer, the References section was completed with the following publications:

Elhakeem, A., Frysz, M., Tilling, K., Tobias, J.H., Lawlor, D.A. Association between age at puberty and bone accrual from 10 to 25 years of age. JAMA Network Open 2019, 2(8), e198918-e198918.

Herbert, A.J., Williams, A.G., Hennis, P.J., Erskine, R.M., Sale, C., Day, S.H., Stebbings, G.K. The interactions of physical activity, exercise and genetics and their associations with bone mineral density: implications for injury risk in elite athletes. Eur J Appl Physiol 2019, 119(1), 29–47. doi: https://doi.org/10.1007/s00421-018-4007-8

Larsen, M.N., Nielsen, C.M., Helge, E.W., Madsen, M., Manniche, V., Hansen, L., Hansen, P.R., Bangsbo, J., Krustrup, P. Positive effects on bone mineralisation and muscular fitness after 10 months of intense school-based physical training for children aged 8–10 years: the FIT FIRST randomised controlled trial. Brit J Sports Med 2018, 52(4), 254–260. doi: http://dx.doi.org/10.1136/bjsports-2016-096219

Soininen, S., Sidoroff, V., Lindi, V., Mahonen, A., Kröger, L., Kröger, H., Jaaskelainenf, J., Atalaya, M., Laaksonenai, D.E., Laitinen, T., Lakka, T.A. Body fat mass, lean body mass and associated biomarkers as determinants of bone mineral density in children 6–8 years of age–The Physical Activity and Nutrition in Children (PANIC) study. Bone 2018, 108, 106–114. doi: https://doi.org/10.1016/j.bone.2018.01.003

Vlachopoulos, D., Barker, A.R., Ubago-Guisado, E., Ortega, F.B., Krustrup, P., Metcalf, B., Pinero, J.C., Ruiz, J.R., Knapp, K.M., Williams, C.A., Moreno, L.A., Gracia-Marco, L.G. The effect of 12-month participation in osteogenic and non-osteogenic sports on bone development in adolescent male athletes. The PRO-BONE study. J Sci Med Sport 2018, 21(4), 404–409. doi: https://doi.org/10.1016/j.jsams.2017.08.018

Tenforde, A.S., Carlson, J.L., Chang, A., Sainani, K.L., Shultz, R., Kim, J.H., Cutti, P., Golden, N.H., Fredericson, M. Association of the female athlete triad risk assessment stratification to the development of bone stress injuries in collegiate athletes. Am J Sport Med 2017, 45(2), 302–310. doi: https://doi.org/10.1177%2F0363546516676262

Comment 7: Line 35 I suggest you use “systematic sports training” instead of “systematic training for sport”

Response: The manuscript was corrected as the Reviewer suggested.

Comment 8: When using the abbreviation TBLH BMD, introduce it on line 45, and then on line 69-70 use the shortcut itself, without the full name.

Response: The manuscript was corrected as the Reviewer suggested.

Comment 9: Line 58 double “were” – correct the sentence. Try to not use the word “subjects”, participants sound better.

Response: The manuscript was corrected as the Reviewer suggested.
